# *Coming Home* (2014) and Its Symptoms

## Jun Lu 

Department of Social Science, York University, Toronto, ON M3J 1P3, Canada; junlu@yorku.ca

**Abstract:** This paper is an in-depth analysis of the Chinese movie *Coming Home* (2014), which mimics the way trauma works and brings the problem of memory into focus. I draw on a psychoanalytic perspective to interpret the storyline, characters, and metaphoric meaning embedded in the construction of the film. My analysis focuses on three symptoms displayed: forgetting, repetition, and historical void. As the most successful Cultural Revolution-related film in the Chinese-speaking world, *Coming Home* confronts the phenomenon of cultural amnesia and visualizes the subjective experience of struggling to remember.

**Keywords:** trauma; symptom; film analysis; psychoanalytic theory; the Chinese Cultural Revolution



## 1. Introduction

16 May 2014, the 48th anniversary of the start of the Cultural Revolution, was marked by the release of *Coming Home* (2014)[1]. This movie depicts a nuclear family dealing with their traumatic experiences after the Cultural Revolution. It is based on Chinese American writer Yan Geling's novel, *The Criminal Lu Yanshi* (Yan 2011), which is considered to be in the style of Scar Literature. Directed by the internationally acclaimed Zhang Yimou[2], *Coming Home* reconfigures the novel by transforming it into a story of aftermath rather than representing the difficult history. Its lack of historical representation contrasts sharply with the novel's abundance of historical detail. Zhang used this narrative technique on purpose, referring to it as "liubai", or leaving a blank. Considering this film as a text, this paper is a reading in the sense of a rescanning of the film. I will not only examine the story told in the movie and how it is told, but also look into what is unsaid but has 'already' been said, what Louis Althusser would call "the internal shadows of exclusion" (du Cinema 1972, p. 8). This analysis will employ psychoanalysis and trauma theory to examine the plot, characters, and underlying implications inherent in the structure and narrative, with a focus on three specific traumatic symptoms: "as-if-forgetting", "repetition", and "historical void". It aims to provide insight into the concept of forgetting as a unique form of remembering, the phenomenon of compulsory repetition as an unspoken expression, and the lack of societal response/witness to traumatic subjects that contribute to the historical void. My inquiry centers on the significance of the film's presentation of trauma in a traumatic manner. How does its inability to depict the historical event illustrate the gap between history as it occurred and history as we remember/receive it? Moreover, how does this film address the pressing question of how collective trauma is treated within the Chinese cultural framework?

## 2. Trauma in the Modern Chinese Context

"Scar" or "wound" is the closest term to trauma for the Chinese to make sense of the aftermath of a difficult history. In the late 1970s, Scar Literature and Introspection Literature emerged as genres of Chinese literature after the end of the Cultural Revolution, which brought to light traumatic experiences and historical reflections. Whereas Scar Literature "often appears to be a mere cathartic expression of individual grievances" (Li and Tam 2019, p. 443), Introspection Literature critiques China's political and social failings and gives

thoughtful consideration to the cultural underpinnings of these issues, frequently reaching far beyond the immediate historical context of the period (Li and Tam 2019, p. 444). However, the discussion on trauma and social and political reflections brought on by the Cultural Revolution suffered a severe setback after the Tiananmen Square Democracy Movement in 1989. To prevent common people from challenging its legitimacy, the Communist leadership is convinced that it must maintain its monopoly on historical interpretation at all costs. Since 1989, "patriotic education" has promoted the idea that the Communist Party has saved China from Western and Japanese humiliation, leading to the rediscovery of the Nanjing Massacre as a national trauma (Beja 2010). However, public discourse becomes impossible when discussing the Cultural Revolution and the Tiananmen Square Democracy Movement. Not only do many people not discuss this period of history, but they also do not discuss the trauma caused by difficult historical periods. Unsurprisingly, mainland China has not developed a trauma theory in response to Scar Literature and Introspection Literature.

While it is impossible to discuss and theorize social and historical trauma in the context of mainland China, there has been progress in the field of individual mental health. German experts were the first group to be invited to give Chinese psychiatrists and psychologists lectures and seminars in the late 1980s and early 1990s. Soon after, the first three-year training program—taught by a team of German psychotherapists—was launched in 1997 and became the Sino-German program (Huang and Kirsner 2020). The systematic effort to introduce and develop psychotherapy also brought a psychoanalytic understanding of trauma to a traumatized country (Plaenkers 2014). Moreover, German experts' reflections on German history and their application of the psychoanalytic framework to work through social and historical trauma inspired their Chinese trainees (Gerlach 2021) and brought stormy classroom discussions on the Chinese situation. Since the early 2000s, mainland China has experienced the emergence of 'psycho-boom,' which refers to "a surge of popular interest in psychotherapy as well as the infiltration of related ideas and values into the cultural sphere" (Huang and Kirsner 2020, p. 8). Following the Wenchuan earthquake in 2008, psychic trauma started to be widely introduced to ordinary people and the Chinese government strives to provide psychological assistance for victims and professional training for mental health practitioners. The predominant viewpoint within the field of mental health suggests that it is imperative to provide Chinese mental health professionals with the necessary training to recognize and address trauma and its psychological ramifications within both cultural and individual contexts. Social and historical trauma remains a sensitive subject, as evidenced by its continued taboo status (Scharff 2014, p. 56).

This shift indicates that the idea of endurance of individual suffering for the collective good in the Chinese mentality began to loosen in the wake of China's adoption of economic reforms and opening-up policies, which ushered in a market-oriented economy beginning in the 1980s. In situations where societal changes require individuals to take responsibility for their actions, examination of the personal experiences of a particular individual can be valuable for comprehension and consideration. Despite the absence of social containment or public hearings over mass individual trauma resulting from political movements, the experiential reality of being wounded remains undeniable. Although the introduction of the concept of trauma to China occurred relatively recently, the lived experience of trauma has remained a persistent phenomenon. Thus, this study focuses on the Chinese experience and examines the portrayal of trauma through a cinematic demonstration.

### 3. Why *Coming Home*

Films depicting the history of the Cultural Revolution have been banned or censored in China. However, because of its stunning cast, acclaimed director, and renowned author of the original novel, *Coming Home* received considerable attention from the start. The marketing strategy of promoting the film as a story about family and love attracted audiences of different ages and brought the film in line with the dominant ideology, allowing it to bypass censorship and gain official recognition. Coupled with commercial operations, it



earned nearly 300 million yuan at the domestic box office, making it the highest-grossing Chinese melodrama film to date (People.cn 2014). As a result, *Coming Home* became the most successful film about the Cultural Revolution, appealing to the largest audiences in mainland China. When *Red Amnesia* (2014)[3], another film about the aftereffects of the Cultural Revolution and its transgenerational effects, was released a year later, it did not receive the same official support as *Coming Home* and failed at the box office. Therefore, *Coming Home*'s ability to reach huge audiences across ages is the first reason I chose it as my object of analysis.

Second, *Coming Home* not only tells a story about trauma, but also unfolds in a traumatized way that showcases the struggles and challenges of representing the past, loss, and forgetting. When he brought *Coming Home* to the 2014 Toronto International Film Festival, director Zhang Yimou voiced his concern over the forgetting of the past. He acknowledged that he was using films to inform and influence people, particularly China's younger generation, about history (Koepke 2015). *Coming Home* serves as a placeholder for audiences to reflect on their own experiences and reclaim the traumatic past by focusing on the suffering of individuals. As Susannah Radstone (2001) points out, "[t]he part trauma film may play in their spectators' integration of individual and collective traumas may therefore mitigate traumatized isolation and create empathy with the sufferings of other in present" (p. 192).

In addition, I consider *Coming Home* to be an example of a modernist trauma film, which aims to expand the possibilities for representing the past and allowing for mourning, remembrance, and forgetting by rejecting the conventional assumption of accurately depicting what happened. (Radstone 2010). However, due to the underdevelopment of trauma discourse in mainland China, the meaning of this film has not been thoroughly examined and requires further analysis and exploration. Gaining access to a traumatic history, as Cathy Caruth (1995b) reminds us, requires "listening beyond the pathology of individual suffering, to the reality of a history that in its crisis can only be perceived in unassimilable forms" (p. 156). Therefore, an in-depth analysis of *Coming Home* will provide an opportunity to process the trauma of the Cultural Revolution by listening to a family's story.

## 4. As-If-Forgetting

Centering on the problem of memory, *Coming Home* challenges our understanding of how we remember, what forgetting is, and how to investigate the memory traces buried beneath the appearance of amnesia. In the film, the dramatic conflict involves a husband returning home only to find that his wife does not recognize him. The inability to remember creates enormous tension between the two major characters: Lu Yanshi, the husband, and Feng Wanyu, the wife.

The film opens with a tense homecoming scene. Lu was a professor who became a political prisoner and was sent to labour reform[4]. In 1973, he escaped the re-education farm and came home to see his family. However, to show her loyalty to the Communist Party, Lu's daughter Dandan informed on Lu to the government, enabling them to catch Lu when he made an appointment to meet Feng at the train station. That morning, Lu was brutally arrested in front of Feng. This violent separation is the first traumatic scene of the story. In late autumn of 1979, when the newly freed Lu walked into the community where he had lived 20 years previously, it did not occur to him that his wife would not recognize him. For Feng, Lu was a complete stranger. Feng, for his part, felt strange to Lu.

However, we should ask whether Feng really forgot her husband. Feng's apartment was covered with various notes reminding herself to leave the door open for Lu to come back. She resented her daughter and blamed her for betraying Lu, her father. Her discontent towards Dandan served as her way of caring for Lu. She clearly remembered going to the train station to pick up her husband on the fifth of every month, dressed appropriately. For the sign that she made in order to find Lu easily in the crowd, Feng practiced her calligraphy by repeatedly writing his name, to ensure she presented the best to him. During his incarceration, she cherished his piano and letters and kept them close to her heart. She

invested a great deal of time listening to his letters and being with him through his writing. When Lu wrote to her and asked her to forgive their daughter, she gladly obeyed and let Dandan move back home. It is obvious that Lu had always been Feng's dear husband in her mind. However, one and half years before Lu's release, Feng started to demonstrate psychogenic amnesia. Thus, the mysterious question at the heart of the film is what is holding Feng back from recognizing Lu and embracing his return.

In the film, the doctor's diagnosis of psychogenic amnesia for Feng implies that the underlying cause of Feng's forgetfulness is psychological trauma rather than physical injury. According to McKay and Kopelman (2009), psychogenic amnesia is a psychological condition characterized by the inability to remember important personal information, often stemming from traumatic or stressful experiences. The individual exhibits a level of memory impairment that surpasses typical forgetfulness and is not attributable to substance misuse or any underlying medical disorder. In the psychoanalytic understanding, psychical trauma contains at least three elements: a violent shock, a wound, and consequences affecting the whole mental organization (Laplanche and Pontalis 1988, p. 466). A very straightforward explanation provided in the movie is Feng Wanyu's experience of a violent shock when, at the train station, she saw not only that her loving husband had been living a subhuman life, but he was also brutally taken away by officials. She was hit on the head and deeply hurt in the heart. The concussion may be the cause, but I argue that Lu's second homecoming is the main trigger of Feng's symptoms.

As a historical movie, *Coming Home* grounds its storytelling in the specific period of Lu's escape from the labour camp in 1973 and his homecoming in late 1979. Feng began to suffer psychogenic amnesia in the middle of 1978 when the Chinese central government issued Central Document No. 11 as the guideline for "removing the rightist hats" (Veg 2014, p. 515) of most of the political rightists like Lu and releasing them from labour camps. Feng would have known that her husband was coming home again and the violent scenes that had occurred a few years before at the train station would have been aroused.

However, the manifestation of trauma does not follow a linear pattern. The phenomenon of experiencing difficulties in memory recall serves as a prominent exemplification of the theoretical concept referred to as "afterwardness" within psychoanalytic discourse. This construct, originating from the German term "Nachträglichkeit" and developed by Sigmund Freud, denotes a retrospective process wherein traumatic significance is attributed to past events in a delayed manner (Sutton 2004). During this process, "earlier memories are re-worked in accordance with later experiences and circumstances" (Aron 2016, p. 273n). In the movie, Lu Yanshi tried very hard to restore his wife's memory by arranging familiar scenes for her: encountering her at the train station dressed in the decent clothes of the good old days; retrieving an old photo showing his handsome face; and playing a beautiful and moving piano piece for her. Unfortunately, Feng still failed to register his appearance. *Memory, History, Forgetting* (Ricoeur 2004) by Paul Ricoeur provides a comprehensive review and profound analysis of how memory functions in our lives and how its presence influences our relationship with the past. According to Ricoeur (2004), Feng's situation can be understood as one in which certain impediments hinder her ability to access her buried memories (p. 444). Even when someone from the Party came to vouch for Lu's identity, she still did not believe that the person in front of her was her husband. Lu Yanshi never considered that he would have to live out the meaning of his name: Yanshi (焉识)[5], "Why could you (not) recognize me?"

The narrative of *Coming Home* compels us to consider the circumstances underlying the as-if-forgetting. Every time Lu approached Feng's bedroom and bed, she screamed at him as if he were Officer Fang, a member of the local Revolutionary Committee during the Cultural Revolution. This phenomenon elicits an unusual feeling in observers, prompting further inquiry: Why Official Fang? The implication of Officer Fang as an intruder in the private realm reveals that Officer Fang had violated Feng Wanyu. It turns out that following Lu's initial homecoming, Feng approached Officer Fang and requested a favour to aid Lu with evading capital punishment for his act of escape. Thus, it is not hard to

understand why Feng may feel shame for having been sexually violated by Officer Fang and for no longer being chaste for her husband. She was terrified that Lu would discover what had transpired and the sacrifices she had made to save him. Consequently, Feng Wanyu cannot recognize Lu as her husband unless she freely reconnects with Lu without repressing her strong and ambivalent emotions. As Clara Mucci (2013) elaborates on the situation of "knowing and not knowing" in traumatized subject, "the reconstruction of the 'real fact' cannot take place without a reconstruction also of the emotional link within the relationship with another being allowing the integration of the affects and the meanings connected to the event that has been traumatic" (p. 73).

Furthermore, Feng's emotional situation was complicated by her conflicting feelings of love and hatred. She had been raising the family and caring for their daughter on her own for twenty years, during which she had to contend with the difficulties and disgrace that Lu had brought upon them. After Lu escaped from the labour camp, Feng was so worried that Lu might get the death sentence that she put herself in a vulnerable place for Officer Fang to take advantage of her. Hence, how could she not feel any resentment for experiencing such difficulties? However, Feng did not express any other emotions besides love when mentioning her husband. She refrained from harbouring resentment and hatred toward Lu because she understood cognitively that her humiliation, disgrace, and hardship were not her husband's fault. This is evident in the scene where she shed tears when she saw the official document that removed Lu's "political rightist" label. However, these conflicting effects can be read into Feng's mistaking of Lu for Officer Fang: spilling out the truth by naming the perpetrator. Feng had therefore made a devastating accusation against Lu by identifying him as the perpetrator, revealing her unconscious resentment towards Lu, whom she believed unconsciously had contributed to her suffering.

Moreover, when Officer Fang assaulted Feng, he was in a position that represented the Party and its power and authority. For Feng Wanyu, the Party's representative, and by extension the Party itself, was not only untrustworthy but also persecutive. She feared being punished like Lu and numerous others who expressed discontent or revealed the truth at the time. It makes sense that Feng claimed to trust the Party when questioned by a Party representative, but she insisted that the man in front of her was not her husband but Officer Fang. Therefore, one can see that Feng Wanyu's traumatic experience includes not only the violent separation at the train station but also the traumatic situations that gave rise to her symptoms.

The symptom of as-if-forgetting in *Coming Home* demonstrates the "dialectics of remembering and forgetting" (Cattell and Climo 2002, p. 1). Through not being able to recognize Lu, Feng's forgetfulness becomes a defensive way of remembering her husband. It creates an ironic distance between the urge to get close to her husband by reading his letters and waiting to see him, and the fact that Lu is readily at hand. Through this ironic distance, Feng Wanyu expressed her internal conflicts between love, hate, fear, desire, and self-preservation. Here, the film effectively illustrates that whereas remembering is an act of linking, forgetting or not knowing are matters of not being able to link or put memories that are thematically and effectively related into a coherent story (Cavell 2006, p. 45). Such a lacuna leads us to the discussion of the problem of the historical void.

## 5. Endless Repetition

Before delving into the meaning of the lacuna and how to comprehend the void of history, I need to investigate Feng's confusing behaviour of waiting at the train station by engaging with psychoanalytic theory on transference and repetition. As demonstrated in the previous section, it is more likely that Feng Wanyu has a memory block than that she has lost her memory. She denied herself access to the ambivalent effects of the traumatic situations. The blocked memory, as Freud discussed in "Remembering, Repeating, and Working Through" (Freud [1914] 1958), speaks in a language of acting out—repetition. In doing so, an individual acts out what has been resisted in order to remember and communicate. The compulsion to express without words outlines a traumatic condition in

which some early experiences can neither be remembered nor forgotten (Roughton 1993, p. 444). Thus, *Coming Home* poses a compelling question to the audience: What does the repetition convey?

After receiving Lu's letter saying that he would come home by train on the fifth, Feng Wanyu went to the train station on the fifth day of every month. Feng's repeated waiting at the station appears to be the result of the letter's incomplete information and her forgetting Lu's face. However, more than that, the repetition is her way of acting out the traumatic memory of Lu's homecoming. In this case, repetition replaces the act of remembering and expresses the memory of real situations, which are repressed in the public sphere through the use of behaviours. It is nonverbal communication that contains meaningful material for understanding Feng's experience of trauma.

As discussed in the previous section, Lu's second homecoming is a situational trigger that activates or intensifies Feng's internal conflicts. Psychoanalytically, this kind of situation is termed transference, during which, as Freud ([1905] 1953) noted, "a whole series of psychological experiences are revived" (p. 116). Melanie Klein (1952) further explained that transference signified the "total situations transferred from the past into the present, as well as of emotions, defences, and object-relations" (p. 437). By interpreting events occurring in a particular present context, transference offers insight into potential occurrences in a comparable past situation. Thus, analysing Feng's repetition via a transference lens helps us understand the 'here and now' as the afterwardness of the 'there and then.'

In Beyond the Pleasure Principle (Freud [1920] 1955), Freud took a child's game as an example to formulate the relationship between trauma and compulsory repetition. In his observation, the child repeatedly threw a wooden reel out and reeled it back using a string. Freud interpreted that the disappearance and return of the wooden reel mimicked the mother's departure and return. By playing this game repetitively, the child was retrospectively trying to gain control of the unpleasant experience of the mother's departure. Similarly, traumatized people are more likely to revisit the traumatic situation and unconsciously attempt to master the experience retrospectively. In *Coming Home*, Feng's repetition is her way of mastering the traumatic separation. She not only blamed her daughter for being the informer but also herself for not opening the door for her husband at his first homecoming and not being able to assist in his escape or ensure his safety. Furthermore, in the film, the missing scenes of Lu's first departure—how he became a political prisoner and was taken away from home—also form part of Feng's mimic game of departure and return. Feng experienced at least two shocks: her husband was taken away suddenly, and then, after suddenly coming back, he was taken away again. Therefore, the repetition of going to the train station and not being able to bring Lu home invites the audience to imagine what Feng experienced years ago, and what kind of experience compels one to express effects in such a pathological way.

According to André Green (2002), in addition to gaining control of traumatic experiences, repetition also serves as a wish-fulfillment through which the individual gains enjoyment or makes things come true (p. 80). What has been repeated is not only the behaviour but also the embedded effects. Every time she repeats the journey, Feng Wanyu walks out of the house with great anticipation and comes back with great disappointment. On one hand, she may attempt to master her feeling of heartbreak; on the other hand, she indulges herself in great hope. Because she is no longer forced to declare a break with Lu and denounce him as a political enemy, her desire for her husband is legitimized. She can finally welcome her husband home and express her love openly. However, she will never "meet" the husband because she "yields to the compulsion to repeat which now replaces the impulsion to remember" (Freud [1914] 1958, p. 151). Such compulsion expresses a force of putting oneself in similar or identical conditions to reproduce effectively (Green 2002, p. 83). Hence, as shown in the movie, no matter how old Feng Wanyu was or how bad the weather was, she never asked anyone else to go to the train station on her behalf.

Moreover, Green (2002) elaborates that the repetition functions as a closed circuit in which a set of interdependent elements form a meaningful whole, and all evolution is

erased by the return of what is repeated (p. 84). In *Coming Home*, Lu Yanshi attempted to break the closed circuit by bridging the memory gap in Feng. However, he gave up after visiting Officer Fang's family. Lu intended to argue with Officer Fang and demand justice for his wife at his residence. Yet, when confronted with Official Fang's wife, who was also furious that her husband had been taken away, Lu realized he did not want to make her life as difficult as Official Fang had made his wife's life. Lu could also imagine how his wife, as a family member of a politically incorrect man, would have endured countless hardships. Lu may have come to the realization that his capacity to seek retribution was hindered by the absence of suitable outlets or individuals to channel his anger towards. He assumed complete responsibility for the suffering experienced by Feng Wanyu. These feelings of incompetence and powerlessness overwhelmed Lu and made him sick after he returned from Official Fang's home. He may have felt foolish for believing that everything would be fine and repaired as long as he could return home. Restoring Feng's memory could be extremely painful and cruel to her. Here, the movie confronts audiences with an ironic and sad scene in which the victimized forgiver, Lu, is left silent while the forgiven, represented by Officer Fang's wife, not only fails to reflect and repent but instead becomes a victim of resentment. For people like Lu Yanshi, how does one fight against the displacement of memory and the wrongdoing of a country and an era?

Before *Coming Home* and its protagonist Feng Wanyu, there was another well-known literary figure in modern Chinese literature who embodied the traumatized woman acting out her repetition. Portrayed by the most significant Chinese modern writer Lu Xun[6], Xianglin's Wife represents the objectification and oppression of women under traditional social norms. She had two arranged marriages, the first after being sold by her parents, and the second by her mother-in-law. She resisted vigorously, but conventional society muffled her voice. After losing her second husband and then her only son, Xianglin's Wife was overwhelmed by guilt and began to seek empathy by repeatedly recounting the traumatic scenes of her son being devoured by animals. Unlike Feng in *Coming Home*, who acts as if no one noticed her strange behaviour, the villagers initially found Xianglin's Wife's repetition to be amusing, though this amusement eventually gave way to disgust, including the I-narrator in the novel. Xianglin's Wife's compulsion to tell never led to greater comprehension but was instead viewed as psychotic and unsettling. No one attempted or was able to help her; only bystanders were present. As a Chinese prototype of a repeated act, the figure of Xianglin's Wife later evolved into a derogatory representation of expressing one's suffering in contemporary culture. Lu Xun's portrayal of this character is highly impactful in illustrating the dilemma faced by individuals who have experienced trauma within Chinese society. The depiction effectively highlights how the desire to articulate their traumatic experiences is overshadowed by their fear of alienation and rejection.

In the original novel, Feng Wanyu repetitively moved around furniture at night. Her neighbours were very angry and vehemently complained about the disturbance. In contrast, *Coming Home* rewrites the character's compulsory repetition into a quiet and unobtrusive routine, as if not being seen by those around her. *Coming Home* foregrounds the memory issue and the symptoms of trauma by removing the response from other people, which suspends social judgment toward the traumatized subject. Repetition ends up being a family's sole burden. Failing to break the closed circuit, the whole family becomes collaborators in the repeated act. At the end of the film, accompanied by the subtitle "many years later", Lu's family starts a day of going to the train station as usual, without saying any words. The silence between family members, the natural station routine, and the subtitle "many years later" create an endless, timeless, and spaceless sense of repetition, as well as a suffocating sense of hopelessness. When the iron gate of the train station closes in front of them again, with the camera shooting from the other side, Feng and Lu appear to be imprisoned. This symbolic final shot sends a strong message that every isolated traumatized subject is trapped in the past, a history that will never pass.

Lu Yanshi's failure to repair Feng's memory, as well as Dandan's silence on her mother's drama, is predetermined. In a society that turns a blind eye to the aftermath of traumatic

history, victims of massive trauma have no way out. The absence of witnesses in the movie, on the other hand, brings out the critical problem of lacking a 'cultural third' (Gerson 2009) to recognize individual and collective trauma. This leads to an in-depth discussion of the historical void in the next section.

## 6. The Problem of Historical Void

The tension between 'the forgotten past' and 'the disrupted present' is well developed in *Coming Home* through Feng's as-if-forgetting and compulsion to repeat. More importantly, the symptom of as-if-forgetting also hangs over the whole narration, as if there was no recollection of the historical details in the film. This adaptation, termed "liubai" or "leaving space or blank", becomes the most contentious aspect of *Coming Home*. In comparison to the original novel, the film's retelling of the story is merely the tip of the iceberg. Many critics note that this is a compromise strategy for Chinese filmmakers who are constricted by censors, particularly when producing a film about the Cultural Revolution. However, there is still strong disappointment in Zhang's decision not to bring history to the screen. For example, Australia-based scholar Cai (2015) refers to this as a "collaboration between Zhang and the coercive powers of both official, mainstream, and commercial discourses as they conspire to intentionally shun, camouflage, depoliticize, and romanticize the painful memories of the socialist revolutionary eras and their subsequent repercussions" (p. 278). She asserts that this reconfiguration turns the story into "a poignant twilight romance" which lacks analytical and mediating power (Cai 2015, p. 288). I, however, make the opposing case. In this section, I will discuss how to understand the problem of the historical void in relation to trauma and what we can learn from the discontent about *Coming Home*'s historical void.

First and foremost, *Coming Home* should be considered an independent story in which a traumatized family is burdened with endless repetition and stuck in the past. I have demonstrated that what appears to be "eternal and innocent love" (Cai 2015, p. 283) is actually hopeless sorrow; without a social support network, there is no way for individuals to escape the lifeless cycle. Indeed, Cai's analysis highlights the negotiation between *Coming Home* and the political conditions of the society that produced it. However, we should not be satisfied with what the film claims to say or what it appears to say; instead, we should focus on the unspoken premises of the narrative. According to media scholar Richard Dyer (2002), the concept of structuring absence should not be understood as the mere absence of elements within a text, nor should it be interpreted as the critic's subjective expectations of what should be present in the text. Instead, the absence is "an issue, or even a set of facts or an argument, that a text cannot ignore, but which it deliberately skirts round or otherwise avoids, thus creating the biggest 'holes' in the text, fatally revealingly misshaping the organic whole assembled with such craft" (p. 83). For the ideological criticism of films, the most important reference factor is not the time frame when the story in the film took place, but the time when the film is produced, distributed, and screened. In this sense, *Coming Home*'s omission of what it was like to experience the Cultural Revolution necessitates further examination and is not merely a compromise with censorial authorities.

The historical void in *Coming Home* is manifested in different aspects. First, in general, it lacks representations of what happened during the Cultural Revolution. Dandan's desire to play the leading role in the ballet The Red Detachment of Women[7] is one of the few scenes representing the fervent competition between young individuals to embody the revolutionary ethos instead of representing the cult of Mao or the violence and destructiveness of class struggle. The movie only uses symbols, such as the printed slogans, the badges of Chairman Mao, the Little Red Book, or the revolutionary song "Sailing the Seas Depends on the Helmsman" to outline the period. Second, as I have discussed, Feng's experiences of the Cultural Revolution are not shown in the film but are buried with strong effects. Third, as a victim of the political movement, Lu Yanshi's experience of being in the labour camp leaves no trace on him and his body. He lost his freedom but came back without any resentment or disciplined behaviours. The same erasure also happens to Dandan, leaving

her internal trajectory of changing from idealization to disillusionment empty. It is not hard to understand that the trauma belongs not only to Feng but to the whole Lu family. These 'holes' and lacuna are there to confront audiences with a question: Do they really have nothing behind these holes? Therefore, the questions "do you really have nothing in relation to this history" and "do people who live through that era really have nothing to it" are implanted in the minds of audiences.

At an individual level, a void or hole is a metaphor for experiential states such as inner deadness or being unavailable to oneself or others due to trauma (Erlich 2003). As historian Dominick LaCapra writes in *Writing History, Writing Trauma* (LaCapra 2001), "[t]rauma is a disruptive experience that disarticulates the self and creates holes in existence; it has belated effects that are controlled only with difficulty and perhaps never fully mastered" (p. 41). Feng Wanyu's amnesia is indeed individual but also reflects the collective condition of the Chinese memory of the Cultural Revolution. This message is implied by Feng's name Wanyu (婉瑜) which is a homophone to 婉喻, meaning to make an implicit analogy. Rather than historical fact, the narrative structure and the characters' conflicts revolve around the difficulty of retelling a story from the past. A nation, as well as an individual, cannot bear to face a particular period of history and chooses an attitude of escape. Thus, analogously, the structural void in *Coming Home* parallels Chinese society's refusal to confront its traumatic past, which can extend beyond the ten years of the Cultural Revolution to include the entire Mao era.

Massive trauma, as elaborated by Laub and Auerhahn (1993), produces a rupture or void that refers to the inability to represent traumatic experiences due to "the collapse of the imaginative capacity to visualize atrocity" (p. 288). This collapse is bridged in the film by focusing on an ordinary family and presenting the human conditions of being in the afterwardness. With this absence, *Coming Home* invites audiences to recall their own experiences of deep and unspeakable effects on family life and to project their imagination to grasp, if only a little, what it might have felt like to be a part of that history. This 'leaving blank' also signifies that the trauma caused by the Cultural Revolution is heterogeneous and can be a series of events and culminates in a set of circumstances. Tomas Plaenkers (2014), a pioneer of psychoanalytic research on Chinese trauma, points out that "[t]he Cultural Revolution was not a homogeneous event that affected an entire society in the same way. On the other hand, a large majority of the population in China did partake of the trauma of the period, yet in different ways" (p. 40). Therefore, despite not depicting the Cultural Revolution and its related movements, the movie does show the pain and near irreversibility of the scars a traumatic past leaves on people and families. In this case, the emotional quality of being wounded resonates with audiences and elicits the recognition of trauma and reflection on our understanding of history.

Moreover, the absence of witnesses in the film is crucial. Lu's family lives in a community, has neighbours, and interacts with people. However, there appears to be no 'witness' to their uncanny situation. A witness, according to Samuel Gerson (2009), can be seen as a 'live third' "whose engaged reaction and concerned responsiveness to the individual's experience create livable meaning" (p. 1343). It can be an individual, relationship, or institution that solidifies the sense of continuity and serves the functions of containing and meaning-making for traumatized people. However, when this support is absent, "the living thirds in which the person was nested now become a nest of dead thirds from which he or she cannot escape" (p. 1343). The absence of social support and the lack of response from others towards this peculiar family renders Lu's family existence akin to that of a desolate island. What *Coming Home* has shown is not only how a family lost its aliveness in close-circuit repetition, but also in a nest of dead social and cultural thirds that turn a blind eye to the existential experience of trauma and the demand for recognition and reparation. Werner Bohleber (2012), who studies trauma and its social impact, describes how traumatized groups require a cultural third, and how its absence would make traumatized individuals much more vulnerable. He says:

Man-made disasters,—such as Holocaust, war, and political persecution—involve specific means of dehumanisation and personality deconstruction, the aim of which is to annihilate the victim's historical and social existence. It is beyond the individual's capacities to integrate such traumatic experiences in a purely personal narrative context; a social discourse is also required respecting the historical truth of the traumatic events, as well as their denial and defensive repudiation. Generally, only scientific explanation and social recognition of causation and guilt are capable of restoring the interpersonal context, thus opening up the possibility to determine what actually happened at the time in an uncensored way. If defensive impulses predominate in society, or rules of silence prevail, traumatized survivors are left alone with their experiences. Instead of drawing support from other people's understanding, they are often dominated by their own guilt feelings, which they come to rely upon as an explanatory principle. (Bohleber 2012, p. 364).

In this vein, we can now comprehend Feng Wanyu's distrust of the Party, Lu Yanshi's giving up on restoring his wife's memory, and Dandan's abandonment of her specialty and becoming an ordinary worker. All these demonstrate what the dead third does to vulnerable traumatized subjects. Feng was compelled to endure a prolonged period of anticipation for her husband's arrival and was unable to discern his identity due to the overwhelming burden of her guilt. Similarly, Lu was compelled to come to terms with the reality of a household that he was unable to return to. Outside of the cinema, public discussions about the Cultural Revolution or other political traumas in Chinese society remain notably absent. In places where we cannot see them, there are countless families trapped in silence, embodying traumatic experiences. The absence of a witness gives a potential answer to the question of why director Zhang's failure to bring historical truth to the screen is so disappointing: the strong demand for a live third.

Early in his career, Zhang Yimou worked diligently to represent history via film. His film *To Live* (1994), which was denied a theatrical release in mainland China, depicts the plights of ordinary people from the 1940s to the Cultural Revolution. While Zhang achieved international acclaim and government recognition in China, he was expected to break the silence of traumatic history with his artistic works. However, when *Coming Home* passed the censorship authorities and was screened on the 48th anniversary of the Cultural Revolution, he received harsh criticism from Dianying Shijie (Cinema World), one of the most prestigious and oldest film magazines in China. In an article titled "Named as the teacher of state, he (Zhang Yimou) is actually an obedient citizen" (Hong 2014, p. 23), the leading contributor of Dianying Shijie (Cinema World) viewed Zhang's transition from self-censorship to self-castration in *Coming Home*. The author asserted that what Zhang called an artistic approach was nothing but rewriting the story in an oversimplified and crude manner that turned a blind eye to an unforgettable history (Hong 2014, p. 25). In my view, it is the high expectations of Zhang and the desire for historical accuracy that prevent critics from appreciating the efforts made to represent the aftermath of the traumatic history. According to the screenplay coordinator, Zhou Xiaofeng's recollection, precisely because directly relevant historical images would be censored and cut, "Zhang Yimou proposed, in the early stages of film preparation, that there should be no flashbacks or voice-overs, and only the current situation should be used to introduce the characters' prehistory and historical background. Zhang believes that refraction, symbolism, and metaphor are utilized when a direct presentation is constrained. Although these techniques are difficult to implement because they require the audience's comprehension and imagination to achieve interactional consonance, we must persistently indicate the signposts and attempt to make ambiguity and implicitness not a sign of weakness, but a mode of thought for the audience. Similar to sending ciphers" (Zhou 2015, pp. 264–65, my translation).

One of the contributions of *Coming Home* is visualizing the belatedness of trauma and emphasizing the subjective experience of time. In doing so, it challenges audiences with "the possibility of a history that is no longer straightforwardly referential" (Caruth 1996,

p. 11). This circularity is shown in Feng Wanyu's psychic reality, in which the current situation is interrupted by previous events. As has been shown through analysis, internal circularity like this can continue for years. The embodiment of trauma alters one's perception of time and history in an existential way. As Cathy Caruth (1996) points out, "[f]or history to be a history of trauma means that it is referential precisely to the extent that it is not fully perceived as it occurs; or to put it somewhat differently, that a history can be grasped only in the very inaccessibility of its occurrence" (p. 18). Such inaccessibility of what happened is represented in *Coming Home* as an empty circle that "symbolizes the absence of representation, the rupture of the self, the erasure of memory, and the accompanying sense of void that are the core legacy of massive psychic trauma" (Laub 1998, p. 507). In this sense, through the absence of representation, the film displays not only the historical power of trauma but also trauma as the (missing) testimony to the past.

Indeed, navigating censorship is a challenge in the Chinese film industry. Ruan (2023) contends that the formulation of rules pertaining to film censorship lacks clarity and precision. The absence of well-defined quantitative criteria and explicit guidelines for making such assessments can result in excessive discretion being granted to administrative censorship, thereby introducing a significant degree of subjectivity into the censorship procedure. In general, themes involving significant historical themes, racial or religious issues, or that realistically depict social conflicts, such as *To Live*, will not be approved. Only projects containing no sensitive historical or social information, such as *Coming Home*, may be made public. The dissatisfaction with or criticism of Zhang and *Coming Home* reflects the demand for a social discourse that, as described by Bohleber (2012), respects historical truth and provides scientific explanations and recognition of causation. However, political reality overshadows the potential for considering the historical void as a traumatic symptom. In addition, it is insufficient to comprehend personal and collective trauma and its symptoms in the Chinese context, as collective traumas are less likely to be represented in cultural products due to the censorship system.

### 7. When Can It End?

One of the film's most intriguing unanswered questions is whether and when Feng is able to recognize her husband and ends the repetition. The silent ending, in which Lu holds a sign bearing his own name and waits for himself to emerge from the station, plunges the audience into an abyss of despair. When three 'Lu Yanshi'[8] simultaneously appear on the screen, even those who find the film silly and absurd cannot laugh. The tragedy of the story is magnified by the fact that the sign "Lu Yanshi" is supposed to represent the physical Lu who has returned but instead represents an imaginary Lu whose return is awaited. The real Lu Yanshi, who was unable to gain his wife's recognition and live with her, has been excluded and has become homeless and ghostly. Analogously, Lu's uncanny situation as a foreign body to the family is what traumatic experience is to a traumatized subject, and what the memory of the Cultural Revolution is to the Chinese people. They are constantly present but never fully integrated. *Coming Home* does a masterful job of subtly presenting the haunting legacy of the past: The amnesiac Feng becomes a monument, a reminder of what happened, and an embodiment of the traumatic experience and traumatic remembering when she stands in front of the train station like a timed alarm clock. Standing next to Feng, Lu realizes he will never be able to give her the Lu Yanshi she desires because that person is lost, both for Lu himself and for Feng Wanyu. Lu is perpetually mourning his lost self, conspiring with Feng's repetition and becoming part of the monument. In doing so, this film serves as a poignant reminder that the traumatic history of a broken past cannot be mended and those who suffer from it are unable to move forward and are constantly mourning their loss.

Disguised as a love story, *Coming Home* is actually a film about trauma through which history leaves its traces and calls for re-evaluation. While people expect to see history on the screen, the film invites audiences to listen to voices from the site of trauma. Its way of storytelling urges a non-conventional understanding of history and truth. By showing

the historical enigma betrayed by trauma, *Coming Home* brings out the problem of what Caruth (1995a) called the crisis of truth, which "extends beyond the question of individual cure and asks how we in this era can have access to our own historical experience, to a history that is in its immediacy a crisis to whose truth there is no simple access" (p. 6). This is demonstrated by the obstructiveness of Feng Wanyu's memory, Lu Yanshi's unknown suffering, and the emptiness of Dandan's development. The stories of the three protagonists are not allowed to be told, just like other countless stories. The breakdown between the signifier and the signified is also the breakdown of the sense of history. When the lived experiences of individuals and their memories are muted, it severs one's relationship with history and damages subsequent generations' sense of reality (Bohleber 2017). As Caruth (1995b) noted, trauma "requires integration, both for the sake of testimony and for the sake of cure" (p. 153). The very basis of saying/telling and listening will start the integration when "a traumatic memory becomes a narrative, a story now told, and with all the appropriate feeling, to another person" (Cavell 2006, p. 43). While historicization requires a witness—the live third—at the individual level, it requires a community of seeing and witnessing for testimony to occur at the national level. In this sense, *Coming Home* tells a quiet and small story in a deafening voice.

**Funding:** This research received Canada Social Sciences and Humanities Research Council doctoral fellowship funding. Funding number: 752-2020-2509.

**Data Availability Statement:** Not applicable.

**Conflicts of Interest:** The author declares no conflict of interest.

## Notes

[1]　The Cultural Revolution was one of a series of political movements launched by Mao Zedong. It began with the "16 May Notice" issued on 16 May 1966. *Coming Home* was released on 16 May 2014.

[2]　Zhang Yimou is the first Chinese director to win the top prize at the three most prestigious international film festivals, the first Chinese director to be nominated for an Academy Award for Best Foreign Language Film, and the director of the opening and closing ceremonies for the Beijing Summer and Winter Olympics.

[3]　*Red Amnesia* is a Chinese movie directed by Wang Xiaoshuai. It premiered in Venice on 4 September 2014 and was released in China on 30 April 2015.

[4]　In the original novel, Lu Yanzhi was one of many intellectuals labelled as rightists during the 1957 Anti-Rightist Movement that imprisoned and sent a large number of Chinese intellectuals to farms for labour reform. The Anti-Rightist Movement is regarded as one of the key factors that paved the way for the Cultural Revolution.

[5]　Yanshi (焉识) in Chinese can be interpreted as "how one could/could not know or recognize something", which is presented as a rhetorical question.

[6]　Lu Xun (1881–1936), real name Zhou Shuren, is regarded as one of the pioneers of modern Chinese literature. Xianglin's Wife is a protagonist of Lu Xun's 1924 short story "New Year's Sacrifice". See (Lu 2009).

[7]　*The Red Detachment of Women* was one of the revolutionary model operas used as a political tool to transform the ideology and values of the Chinese people during the Cultural Revolution.

[8]　After completing the draft, I discovered a similar reading of the 'three Lu Yanshi' in Dai Jinhua's lecture on film studies. See (Dai 2018, 1:02:30–1:03:45)

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
