# Peer review of "Coming Home (2014) and Its Symptoms"

_arts, 2014_

Round 1

Reviewer 1 Report

This is an interesting article that invites a reconsideration of Zhang Yimou's Coming Home (2014). The article explores how trauma is dramatised in the film. The author uses a classical psychoanalytical approach to explain the depiction of trauma in the film. This approach is effective in demonstrating the film's use of absence to signal the trauma of the Cultural Revolution.

The article becomes stronger as it goes along and its discussions of how the film navigates censorship, and the mixed critical reception it received on release, are compelling and nicely argued.

The article puts itself into conversation with key foundational names in trauma studies - Caruth, Laub, LaCapra - as well as some more recent scholarship in the field.

The classical psychoanalytical approach chosen leads the author to make an unfounded claim at the start of the article 'it is impossible to discuss and theorize social and historical trauma' (p. 2). As I note in my comments this is simply not true. Moreover, the discussion that ensues in this same paragraph works against this claim. As well as inviting the author to rethink this claim, I would also like to suggest they take a look at those critics and theorists who engage with trauma as a social and historical phenomenon (e.g. Achille Mbembe, and Franz Fanon in a post/colonial context, Kai Erikson in a sociological one). Engagement with this material would help to nuance the claims made in the article at various points. It would also provide the author with some additional context for discussing trauma in the social and historical ways that are pursued in the article.

I have made various comments and suggestions on the phrasing and grammar, which can be found on the pdf attached. Overall the expression is clear but there are quite a few places where phrasing goes awry. This should be tidied up before publication to ensure the expression matches the quality of ideas.

Reviewer 2 Report

I thoroughly enjoyed reading this essay which applies theories of trauma and representation to the Chinese film Coming Home. Overall, the author handles the multiple aspects of the argument well, balancing attention to theories of trauma and representation, many from a western context, with specific analysis of Chinese cultural, historical, and political contexts that shape the film and its reception. 

I have made some notes by line, which I offer here. Overall, they focus on structural issues of revising the introductory section to introduce some of the main concepts, such as "As if forgetting" and "Live third" or witness, both of which are key to the paper. The introduction ends with three questions, but I suggest adding one about a lack of a witness, either interior to the film or outside, in the context of Chinese censorship, which allows the opportunity to introduce the issue of that broader political/historical context as it relates to the arguments. 

The arguments about individual v. collective identity are important--including the idea that the film uses individual experiences as a marker for larger collective ones, and the discussions of repetition and displacement, especially in the form of Feng's waiting at the station, are excellent. 

Overall the essay is an excellent contribution not only to the discussion of Chinese art/literature/film, but also to the literature on trauma and representation, which the author handles skillfully. 

Line notes:

Line 162, develop argument re: psychogenic amnesia with at least one more line — needs a definition and a sense of how it is applied here Line 172 - again, need definition of “afterwardness” as opposed to memory The concept of “As-if forgetting” needs explication—is this the author’s term, a theoretical term from trauma studies? Should be introduced up front as it is key to the article 182 for non Chinese speakers - is this the literal translation of his name? With the (not) or without? 191-210 - interesting reading of Feng’s misreading of Lu as Fang, but needs a stronger explanation of this displacement or projection from Lu’s violator to Lu himself…using psychoanalytic or trauma theory—a strong reading of her “spilling out the truth by naming the perpetrator” — but needs more methodical clarity to help reader understand. In what sense is Lu a "perpetrator?"   233 no need for the direction: “which will come up in a later section;” the transition at the next paragraph works well to demonstrate that  249 how does her waiting at the station result from her forgetfulness? The next line about repetition raises a more compelling theory 260-264 excellent, revelatory use of theory 294-5 could the reason Feng never invites her daughter to go in her place to the station have more to do with her anger at her daughter for turning her father in? 312-313 - terminology a bit confused: is the “victimized forgiver” Feng and the “forgiven” Lu? Or…? 350 Dandan’s silence - could use development; what we mostly know of her is that she spoke to authorities, resulting in the imprisonment of her father; where does her "silence" come in? 352 “absence of otherness” needs explanation; explicated in line 444 but should come earlier, perhaps even in opening/introductory paragraph, as a term of consideration in the paper, especially as it raises the concept of witness, central to theories of trauma and representation 407 “do they really have nothing” is obscure here. Is it a quote from the film or the author’s construction? Does it imply "nothing" as no memories, no traces of the event?  472 remove word “that" 503 replace “presenting” with “presentation” or “representation” 523 suggest: "also trauma as the (missing) testimony to the past" — I add (missing) because of the emphasis on the Director’s lack of direct address, e.g., this is not a direct testimony to the past—another expression of present absence 529 is it not precisely the collective trauma that is missing here? This brief paragraph on the censorship regime could be expanded to give more detail on criteria, how it is that one film is accepted by censors and another not—it seems that Coming Home was accepted precisely because it was not explicit about the traumatic history of the Cultural Revolution, whereas To Live was  541 is Lu actually homeless? Clarify… 546 idea of Lu as a “monument” in train station is fascinating and evocative—if there is room, could be expanded  

In terms of language, the essay needs a careful proofread that addresses grammatical issues (mostly re: articles like "the" or "a" as well as pronoun agreement, as in line 136, which as the sentence is currently written should be "her." Also some instances of syntax, as in line 281, or incomplete thought, as in line 395, Dandan’s desire to be in ballet is “one of the few scenes..." begs the question, "one of the few scenes that what…?" 

Author Response

Dear Reviewer,

I would like to express my gratitude for the considerable time and effort you have invested in offering your invaluable feedback on my manuscript. I deeply value your perceptive observations and constructive comments. I have implemented modifications in order to incorporate the majority of the suggestions provided by you. The modifications made to the manuscript have been indicated with track-changes.

Here is a point-by-point response to the reviewers’ comments and concerns.

Comment 1: “revising the introductory section to introduce some of the main concepts

Response: Thank you for pointing this out. I agree with this comment. I have revised the introduction paragraph to read as follows:

“This analysis will employ psychoanalysis and trauma theory to examine the plot, characters, and underlying implications inherent in the structure and narrative, with a focus on three specific traumatic symptoms: “as-if-forgetting,” “repetition,” and “historical void.” It aims to provide insight into the concept of forgetting as a unique form of remembering, the phenomenon of compulsory repetition as an unspoken expression, and the lack of societal response/witness to traumatic subjects that contributes to the historical void. My inquiry centers on the significance of the film’s presentation of trauma in a traumatic manner. How does its inability to depict the historical event illustrate the gap between history as it occurred and history as we remember/receive it? Moreover, how does this film address the pressing question of how collective trauma is treated within the Chinese cultural framework?”

Comment 2: add definition or explanation for key concepts, such as psychogenic amnesia, afterwardness, as-if-forgetting, and the absence of otherness

Response: I appreciate your pointing these out. Accordingly, I have altered the context to improve comprehension. Please see line 166-173 for psychogenic amnesia, line 189-196 for afterwardness, and line 29 & line 135-137 for what I called ‘as-if-forgetting.’ I changed ‘the absence of otherness’ to “the absence of witness’ to connect it to the  elaboration on witness as a live third, and I also rearranged some sentences for clarity. Please see line 494-507.

Comment 3:  clarify some points

Response: Thank you for pointing these out. I have revised sentences for clarity.

  1. For the meaning of Lu’s name, please see the revised note no. 5.
  2. For “how does her waiting at the station result from her forgetfulness,” please see line 281-283. I have revised this point to read as: “Feng’s repeated waiting at the station appears to be the result of the letter’s incomplete information and her forgetting Lu’s face”
  3. For “could the reason Feng never invites her daughter to go in her place to the station have more to do with her anger at her daughter for turning her father in?”, I have revised this point to read as: “Hence, as shown in the movie, no matter how old Feng Wanyu was or how bad the weather was, she never asked anyone else to go to the train station on her behalf.” (line 333-334)
  4. For “terminology a bit confused: is the ‘victimized forgiver’ Feng and the ‘forgiven’ Lu?”, I have revised this point to read as: “Here the movie confronts audiences with an ironic and sad scene in which the victimized forgiver, Lu, left with silent while the forgiven, represented by Officer Fang’s wife, not only failed to reflect and repent, but instead became a victim of resentment.” (line 357-359)
  5. For “‘do they really have nothing’ is obscure here,” I have revised this point to read as: “These ‘holes’ and lacuna are there to confront audiences with a question: Do they really have nothing behind these holes? Therefore, the questions “do you really have nothing in relation to this history” and “do people who live through that era really have nothing to it” are implanted in the mind of audiences.” (line 458-461)

  1. For “is Lu actually homeless?,” here I was referring to the fact that Lu could not return home to his wife and daughter. I have revised this point to read as: “The real Lu Yanshi, who was unable to gain his wife’s recognition and live with her, has been excluded and has become homeless and ghostly.” (line 608-609)

  1. “In what sense is Lu a ‘perpetrator’?” – Here I mean that Lu can be identified as a significant contributor to the hardship experienced by Feng in relation to his escape and political standing, despite the potential interpretation of Lu’s innocence.

I have revised this point to read as: “Feng had therefore made a devastating accusation against Lu by identifying him as the perpetrator, revealing her unconscious resentment towards Lu, whom she believed unconsciously had contributed to her suffering.” (line 242-245)

  1. “Dandan’s silence” – here I refer to Dandan’s silence to her mother’s repetition drama. I have revised this sentence to read as: “Lu Yanshi’s failure to repair Feng’s memory, as well as Dandan’s silence to her mother’s drama, is predetermined. In a society that turns a blind eye to the aftermath of traumatic history, victims of massive trauma have no way out.” (line 400-402)

I appreciate your inquiry about the character Dandan. Thank you for the recommendation. This aspect would have been intriguing to investigate. However, it may be slightly beyond the scope of this analysis to discuss her psychological trajectory, which the director provided us with little information to trace, and I have indicated this as another layer of emptiness.

Comment 4:  expand some points

Response: I appreciate you thoughts on these points. Accordingly, I have expanded the elaboration to improve comprehension.

  1. For the idea of Feng as “monument,” I have modified this idea to considering both Feng and Lu together. Now it reads as

“Coming Home does a masterful job of subtly presenting the haunting legacy of the past: The amnesiac Feng becomes a monument, a reminder of what happened, and an em-bodiment of the traumatic experience and traumatic remembering when she stands in front of the train station like a timed alarm clock.Coming Home does a masterful job of subtly presenting the haunting legacy of the past: The amnesiac Feng becomes a monument, a reminder of what had happened when she stands in front of the train station like a timed alarm clock. Standing next to Feng, Lu realized he would never be able to provide her with the Lu Yanshi she desired. Because that person had been lost, for both Lu himself and Feng Wanyu. Lu is perpetually mourning his lost self, conspiring with Feng's repetition and becoming part of the monument. In doing so, this film serves as a poignant reminder that the traumatic history a broken past that cannot be mended and those who suffer from it are unable to move forward and are constantly mourning their loss.” (line 614-626)

  1. Regarding on the censorship regime and “how it is that one film is accepted by censors and another not,” I have added a new source and expanded this part to read as below,

Indeed, navigating censorship is a challenge in the Chinese film industry. Ruan Jiale (2023) contends that the formulation of rules pertaining to film censorship lacks clarity and precision. The absence of well-defined quantitative criteria and explicit guidelines for making such assessments can result in excessive discretion being granted to administrative censorship, thereby introducing a significant degree of subjectivity into the censorship procedure. In general, themes involving significant historical themes, racial or religious issues, or that realistically depict social conflicts, such as To Live, will not be approved. Only projects containing no sensitive historical or social information, such as Coming Home, may be made public. The dissatisfaction with or criticism of Zhang and Coming Home reflects the demand for a social discourse which, as described by Bohleber (2012), respects historical truth and provides scientific explanations and recognition of causation. However, political reality overshadows the potential for considering historical void as a traumatic symptom. In addition, it is evidently insufficient to comprehend personal and collective trauma and its symptoms in the Chinese context, as collective traumas are less likely to be represented in cultural products due to the censorship system. (line 582-597)

Comment 5:  suggest: "also trauma as the (missing) testimony to the past" — I add (missing) because of the emphasis on the Director’s lack of direct address, e.g., this is not a direct testimony to the past—another expression of present absence

Response: Thank you for your extremely insightful suggestion. I value your thoughts on this point. I agree completely and have amended the sentence as you suggested.